# Identification of Runs of Homozygosity Islands and Functional Variants in Wenchang Chicken

**DOI:** 10.3390/ani13101645

**Published:** 2023-05-15

**Authors:** Shuaishuai Tian, Wendan Tang, Ziqi Zhong, Ziyi Wang, Xinfeng Xie, Hong Liu, Fuwen Chen, Jiaxin Liu, Yuxin Han, Yao Qin, Zhen Tan, Qian Xiao

**Affiliations:** Hainan Key Laboratory of Tropical Animal Reproduction & Breeding and Epidemic Disease Research, College of Animal Science and Technology, Hainan University, Haikou 570228, China; tianshuaishuai163@163.com (S.T.);

**Keywords:** Wenchang chicken, inbreeding, genetic diversity, runs of homozygosity, selection signatures

## Abstract

**Simple Summary:**

Wenchang chickens are the only chicken breed listed in the ‘animal genetic resources in China (poultry)’ in Hainan Province and are famous for their excellent meat quality. Protection of this genetic resource may ensure poultry husbandry’s sustainable and successful development. For more effective conservation, development, and utilization of this genetic resource, we investigated the diversity, degree of inbreeding, and runs of homozygosity (ROH) patterns for Wenchang chickens using whole-genome sequencing data. Our analysis showed that the genetic diversity of Wenchang chickens was relatively high. Selection signal analysis of Wenchang chickens based on ROH found some candidate genes that were putatively associated with meat quality traits and stress resistance traits, such as disease resistance and heat tolerance.

**Abstract:**

Wenchang chickens, a native breed in the Hainan province of China, are famous for their meat quality and adaptability to tropical conditions. For effective management and conservation, in the present study, we systematically investigated the characteristics of genetic variations and runs of homozygosity (ROH) along the genome using re-sequenced whole-genome sequencing data from 235 Wenchang chickens. A total of 16,511,769 single nucleotide polymorphisms (SNPs) and 53,506 ROH segments were identified in all individuals, and the ROH of Wenchang chicken were mainly composed of short segments (0–1 megabases (Mb)). On average, 5.664% of the genome was located in ROH segments across the Wenchang chicken samples. According to several parameters, the genetic diversity of the Wenchang chicken was relatively high. The average inbreeding coefficient of Wenchang chickens based on *F_HOM_*, *F_GRM_*, and *F_ROH_* was 0.060 ± 0.014, 0.561 ± 0.020, and 0.0566 ± 0.01, respectively. A total of 19 ROH islands containing 393 genes were detected on 9 different autosomes. Some of these genes were putatively associated with growth performance (*AMY1a*), stress resistance (*THEMIS2*, *PIK3C2B*), meat traits (*MBTPS1*, *DLK1*, and *EPS8L2*), and fat deposition (*LANCL2*, *PPARγ*). These findings provide a better understanding of the degree of inbreeding in Wenchang chickens and the hereditary basis of the characteristics shaped under selection. These results are valuable for the future breeding, conservation, and utilization of Wenchang and other chicken breeds.

## 1. Introduction

Wenchang chickens are a typical native broiler breed in China, mainly in Hainan province, the southernmost part of the country. Being the descendants of those lines who have lived a long time under environmental conditions with high humidity and temperature has resulted in Wenchang chickens performing well when undergoing heat stress and exposure to zoonotic diseases. In addition, Wenchang chickens are well known for their excellent meat quality and high prolificacy. Due to their superior meat quality, Wenchang chicken possesses a large share of the poultry market in Hainan province, and its production yields are sold in Southeast Asian countries. However, the introduction of commercial breeds put Wenchang chickens in danger due to their subpar performance in terms of growth traits and feed conversion ratios [1]. As a result, the number of Wenchang chickens has been decreasing in the past few decades, which may lead to occurrences of inbreeding. The deficient control of inbreeding may give rise to weaknesses in the genetic variability and genetic diversity of Wenchang chickens, which have a detrimental effect on conserving this valuable genetic resource and the sustainable development of poultry husbandry. In addition, inbreeding may also increase the probability of genetic drift and the frequency of autozygosity for deleterious alleles, consequently reducing performance for individuals in the Wenchang population. Therefore, to better preserve the genetic diversity and utilization of Wenchang chicken, we must find an effective way to characterize and understand inbreeding and autozygosity in this valuable genetic resource.

Runs of homozygosity (ROH), the continuous homozygous segments in an individual genome, are common in human and animal populations [2]. ROH segments are identical haplotypes transmitted from parents to offspring, so these segments can be hereditary in a population and provide information about the population’s history and demographic evolution. Long ROH are generally due to recent parental relatedness, while shorter ROH indicate more ancient common ancestors in the pedigree [3]. Thus, detecting ROH can estimate the whole genome’s inbreeding level, which can be used to improve mating systems and minimize inbreeding [4]. The assessment of whole-genome inbreeding based on ROH is widely used and effectively distinguishes between recent and ancient inbreeding. In addition, the inbreeding coefficient (F) estimation method by ROH is suitable for large populations [4].

Besides consanguineous mating and population size reduction, selection pressure can also result in long homozygous regions along the genome [5]. Studies have claimed that natural and artificial animal selection has resulted in breeds with extensive phenotype variation [1,6]. Using selection signatures to identify regions in the genome under selective pressure may help us determine harbored genes and variants that modulate important animal phenotypes. Nowadays, more and more researchers perform ROH to reveal genetic mechanisms of important traits, and it has been widely used in a variety of animals, such as cattle [7], pigs [8], and sheep [9]. For example, Li et al. detected ROH in Hu sheep based on sequencing data and identified selected genes within the ROH islands relevant to agricultural economic characteristics [10]. However, these types of studies are seen less commonly in chickens, especially Chinese indigenous chickens such as the Wenchang chicken.

Therefore, the present study aimed to estimate the diversity and detect ROH patterns in Wenchang chicken populations, observe the degree of inbreeding in Wenchang chickens, and identify candidate genes related to breed-specific traits of Wenchang chickens from within ROH islands. The results of this research contribute to our understanding of inbreeding in Wenchang chickens and help elucidate how artificial or natural selection affects the distribution of functional variants at the whole-genome level.

## 2. Materials and Methods

### 2.1. Animals and Genotypes

In this study, 235 individual Wenchang chickens were sampled from three local conservation farms in Hainan province, China (Appendix A). All selected individuals’ genomes were sequenced using the Illumina Nova Seq platform (Illumina, San Diego, CA, USA) and 150-base pair (bp) paired-end sequencing [11]. For details, see Appendix A. Raw data from re-sequencing were filtered using the fastp software (a FASTQ data pre-processing tool) with the default parameters. After filtering, the remaining reads were aligned to the chicken reference genome (bGalGal1.mat.broiler.GRCg7b, https://www.ncbi.nlm.nih.gov/assembly/organism/9031/latest/) (accessed on 1 August 2021) using Burrows–Wheeler Aligner (BWA, version 0.7.17) [12]. GATK4 software (version 4.1.6.0) [13] was used for the single nucleotide polymorphism (SNP) calling of each individual. STITCH [14], a software program that can perform this task using ultra-low coverage data, was used for the imputation of the missing genotypes. In addition, SNPs were filtered using PLINK (v1.9) [15]. SNPs were retained for further analysis using the following criteria: (1) minor allele frequency (MAF) ≥ 0.05; (2) missing rate ≤ 0.1; (3) calling rate ≥ 0.9; (4) the calling quality ≥ 30. After that, the remaining SNPs were used for further downstream analysis.

### 2.2. Genetic Diversity and Linkage Disequilibrium (LD) Analysis

The estimation of the genetic diversity of Wenchang chickens was performed using some indices, including expected heterozygosity (H_E_), observed heterozygosity (H_O_), the percentage of polymorphic loci (*P_N_*), the minor allele frequency (MAF), and nucleotide diversity (*pi*). H_E_, H_O_, *P_N_*, and MAF were calculated using PLINK (v1.9) [15], and *pi* was calculated using VCFTOOLS (version 0.1.16) [16]. The squared correlation (*r*^2^) between pairwise SNPs served as a measure of the linkage disequilibrium (LD) decay using PopLDdecay with default parameters [17].

### 2.3. Identification of ROH

ROH on all autosomes of each individual were identified using PLINK (v1.9) [15] with a sliding window. In addition, according to previous studies [18,19], specific criteria were applied based on the following: (1) Each sliding window should contain 50 SNPs across the genome; (2) due to genotyping error, up to five SNPs with missing genotypes and one SNP with a heterozygous genotype were allowed for each ROH; (3) each ROH should have a sequence of more than 50 consecutive SNPs; and (4) only detect segments with ROH length greater than 100 kilobases (kb). ROH extracted from sequence data were further classified into three length categories: short ROH (<1 megabase (Mb)), medium ROH (1–2 Mb and 2–3 Mb), and long ROH (>3 Mb). In addition, to further understand the influence of different parameters, we use the detection method of adding 50 SNPs at a time to verify the influence of different parameters.

### 2.4. Assessment of Inbreeding Coefficients

Three methods (*F_ROH_*, *F_HOM_*, and *F_GRM_*) were used to estimate the inbreeding coefficients of Wenchang chicken populations. First, the *F_ROH_* was calculated according to the method proposed by McQuillan et al. [20], which was defined as the ratio of the total length of ROH to the total length of the genome covered by the analyzed SNPs or sequences. Formula (1) was used as follows:(1)$$F_{ROH}=\frac{\sum L_{ROH}}{\sum L_{auto}}$$
where ∑*L_ROH_* is the total length of all the ROH for one individual across the genome, and ∑*L_auto_* is the total length of the autosomal genome covered by the analyzed SNPs, which was 900 Mb in our study. This length is consistent with the chromosome length of chickens reported in previous studies [21].

Second, *F_HOM_* was calculated using PLINK (v1.9) [15] to assess the number of observed and expected autosomal homozygous genotypes for each sample [7]. Finally, the genomic inbreeding coefficient of each individual was evaluated from the genomic relation matrix (*F_GRM_*) according to the previous method proposed by VanRaden [22]. *F_GRM_* coefficients were estimated using the option “--ibc” from the GCTA software. The genome relation matrix is obtained and used by us to calculate the diagonal of the matrix to calculate each individual’s *F_GRM_* value. The formula is as follows:(2)$$F_{GRMj}=G_{jj}-1,$$
where *F_GRMj_* is the genome inbreeding coefficient of each individual, and *G_jj_* is the diagonal element of the genome relation matrix [21].

### 2.5. Identification of Candidate Genes within ROH Islands

PLINK (v1.9) was first utilized to identify ROH islands with the command “--homozyg” [23]. Secondly, the frequency of each SNP that appeared in an ROH was calculated, and the percentage of SNPs that existed within an ROH was subsequently estimated. Thirdly, the top 1% of SNPs were defined as candidate SNPs, and the genes underwent further identification. Finally, Gene Ontology (GO) terms and Kyoto Encyclopedia of Genes and Genomes (KEGG) pathways were analyzed to identify the functions of candidate genes further using DAVID (v6.7, https://david.ncifcrf.gov/) (accessed on 26 December 2022) [24]. Only the p-values with less than 0.05 of these enriched GO terms and KEGG pathways were considered significant and listed in this study.

## 3. Results

### 3.1. SNP Identification

After quality control and filtration, 16,511,769 SNPs were obtained from the whole genomes of 235 Wenchang chickens. To further understand the distribution characteristics of these SNPs, we first categorized them into their functional classes. As a result, 73,238 (28.60%) SNPs were categorized as nonsynonymous, and 182,812 (71.40%) SNPs were categorized as synonymous. A total of 4,857,664 (29.30%) SNPs were obtained in intergenic regions. For the SNPs found in gene regions, a total of 257,930 (1.56%) SNPs were in exon regions, and 694,328 (4.19%) SNPs were in untranslated regions (Appendix A). The R package (v4.1.3) “CMplot” then calculated and visualized the density distributions of the SNPs found in Wenchang chicken on each chromosome. This result is depicted in Appendix A. The distribution of SNP has about one SNP site per 17.47 kb. Chromosomes 1 and 2 displayed the largest number of SNPs, with 3,424,555 and 2,584,382, respectively, while chromosome 6 had the highest densities of SNPs. Since a region with a high density of SNPs was contained in chromosome 6, this suggests that chromosome 6 may be an important target for further research. 

### 3.2. Genetic Diversity and LD Analysis

The results of the genetic diversity indices are shown in Table 1. H_E_ (0.24) was found to be slightly higher than H_O_ (0.23). The value of *P_N_* was 0.83, and the *pi* value was 0.0043 (Figure 1A). The average value of MAF was 0.17, varying from 0.01 to 0.50. The MAF of more than 10.67% of the SNPs was higher than 0.40, and the MAF of 43.90% of the SNPs was lower than 0.10 (Appendix A). 

LD analysis was measured with *r*^2^ values and could provide further information on the overall diversity level of the Wenchang chicken population. Overall, the LD value decreased with the increased distance between SNPs, and the decay was rapid in Wenchang chicken (Figure 1B). In brief, these results indicated that most Wenchang chickens displayed high genetic diversity.

### 3.3. Genomic Distribution of ROH

A total of 53,506 ROH segments were identified in 235 Wenchang chickens through ROH detection on all autosomes (Figure 2). The overall mean length of detected ROH was 53.53 ± 9.47 Mb, with a mean number of 227.69 ± 33.65 ROH per animal. The coverage of the identified ROH segments per chromosome ranged from 2.67% to 8.88% in Wenchang chickens. The chromosomes of 33 (61.18%) and 31 (41.15%) had very high ROH coverage. Summary statistics of the numbers of ROH segments across different length classes are shown in Table 2. The average size of each segment was 0.2351 Mb, ranging from 0.10 Mb to 4.74 Mb, and the longest fragment was found in chromosome 2 (which contains 43,925 SNPs). In terms of the identified segments, we can find that the majority (98.17%) of the whole ROH length was made up of the short segments (<1 Mb), while the long ROH segments (>3 Mb) accounted for just 0.03% of the whole ROH length, indicating that ROH covered the highest proportion of the genome (88.53%) in Wenchang chickens. The influence of different detection parameters on the results is shown in Appendix A. With the increase of SNP in a single test, the number of short fragments decreased until the number of short fragments (ROH < 1 Mb) accounted for 98.47%, while the number of long fragments was still 0.03%. 

### 3.4. Inbreeding Coefficients

The results of the inbreeding coefficients calculated by different methods are shown in Table 3. The mean value of *F_ROH_* in the 235 Wenchang chicken sample population was 0.0566, with a range of 0.0267 to 0.0888, and the value of *F_HOM_* ranged from 0.0281 to 0.1527, with a general mean of 0.05614. The values of *F_GXM_* ranged from 0.02168 to 0.12298, with a general mean of 0.05999. The inbreeding coefficient values obtained using the three methods were roughly the same and remained low, indicating that the level of inbreeding in the Wenchang chicken population was relatively low. Furthermore, the inbreeding coefficients estimated based on the different physical lengths of the ROH fragments varied greatly, of which, *F_ROH_* (<1 Mb) was significantly larger than *F_ROH_* (1–2 Mb), *F_ROH_* (2–3 Mb), and *F_ROH_* (>3 Mb). Furthermore, a strong correlation (0.94) was found between *F_ROH_* (<1 Mb) and *F_ROH_* (All), while the weakest correlation (0.15) was found between *F_ROH_* (>3 Mb) and *F_ROH_* (All) (Figure 3). This result indicated that short ROH fragments might play a major role in calculating *F_ROH_*.

### 3.5. ROH-Based Selective Signal Analysis

ROH hotspots or islands were defined as the genomic regions with the highest frequency of ROH occurrence. In total, 19 ROH islands, 77,975 SNPs, and 393 genes under selection were detected across the 39 autosomes based on their occurrence in the top 1% of the ROH islands as possible regions for candidate genes (Figure 4 and Appendix A). The length of these genomic regions ranged from 574.6 bp on Chromosome 2 to 0.9 Mb on Chromosome 1. Notably, the genomic region with a length of 1.7 kb located on Chromosome 8 contained 91 genes, which might be the most relevant region for functional expression in the Wenchang chicken population. 

The functional annotation of the identified genes was analyzed, and the results are depicted in Figure 5. In total, 25 GO terms and one pathway were significantly enriched (Appendix A). GO clustering analysis revealed that the genes that contained SNPs were significantly enriched for metabolic processes (GO:0005975-carbohydrate metabolic process; GO:0006004-fucose metabolic process), activity processes (GO:0004556-alpha-amylase activity; GO:0046922-peptide-O-fucosyltransferase activity), development processes (GO:0035987-endodermal cell differentiation; GO:0098609-cell adhesion), binding processes (GO:0005178-integrin binding; GO:0005518-collagen binding; GO:0000049-tRNA binding; GO:0003725-double-stranded RNA binding); and collagen binding (GO:0005518). Genes containing SNPs were significantly enriched for the KEGG pathway “ECM-receptor interaction (GGA04512)”.

## 4. Discussion

### 4.1. Genetic Diversity of the Wenchang Chicken Sample Population

Numerous factors affect genetic diversity, and understanding genetic diversity is essential for developing conservation and sustainable livestock management programs. Our study used several indices to estimate the genetic diversity of Wenchang chickens. Heterozygosity, including observed heterozygosity (H_O_) and expected heterozygosity (H_E_), one of the most widely used genetic diversity parameters, was first carried out to estimate the Wenchang chicken population. In comparison with other chicken breeds reported in previous research, the heterozygosity (H_O_) and expected heterozygosity (H_E_) values of Wenchang chicken estimated in our study are relatively higher when compared to Italian local chicken breeds (H_O_ = 0.1626 ± 0.200) [23], Swedish chicken breeds (H_O_ = 0.225 ± 0.023) [24], Guangzhou chickens (H_E_ = 0.2114), Huiyang chickens (H_E_ = 0.2376), and Commercial broilers (H_E_ = 0.2337) [25]. In addition, many studies have also extensively used the MAF index to evaluate genetic diversity [26]. The distribution of MAF may provide valuable insights into the distinction between common and rare variants in the population. A higher proportion of one population’s low MAF values may indicate high genetic diversity [27]. In our study, the average value of MAF in Wenchang chickens was 0.17, and the proportion of MAF values less than 0.1 was 43.90%. These results indicated that the genetic diversity level of Wenchang chickens was high, consistent with heterozygosity. Likewise, the pi value in Wenchang chickens was higher than other chicken breeds reported in previous studies, such as Guangzhou chickens (0.00199), Beijing chickens (0.00216) [26], and Dongzhongai chickens (0.00332) [28]. The genome’s high pi value also provides evidence for the fast-decaying LD of Wenchang chickens.

LD, an important genetic phenomenon, is a non-random association of alleles at different loci. The pattern of LD decay between genetic markers can provide valuable views on a population’s history and evolution. Compared to other chicken breeds, the LD extension of the Wenchang chicken population was much smaller than commercial chicken breeds, such as White Leghorn [28], and was at a low level among some Chinese indigenous chicken breeds, such as Xichuan black bone chicken [29], Xianju chickens, and Taihe chickens [26]. The degree of LD decay rapidity indicates high genetic diversity and is associated with processes such as migration, selection, and genetic drift in one population. Commercial chicken breeds have undergone strong artificial selection for economic traits, which may result in a low LD decay rate and diversity decline. Altogether, Wenchang chickens displayed a high level of genetic diversity, which indicated that the protection strategy for Wenchang chickens used on conservation farms was effective. However, a variety of dynamic factors influence genetic diversity. Therefore, genetic variation and structure should be continuously monitored in the future to prevent the rapid decline of diversity, which is significant for the sustainable development of the poultry industry. 

### 4.2. Characteristics of the Identified ROH

Analysis of the distribution of ROH across the genome and the number of ROH classified by their physical lengths can provide valuable information on the genetic history and inbreeding of populations [30]. Due to the occurrence of recombination events, the length of the ROH decreased over time. Hence, longer ROH segments indicate that inbreeding events were recent, while shorter ROH segments were remote [31,32]. In our study, the distribution of ROH segments across the genome of the Wenchang chicken sample population was mostly short segments, while the percentage of long segments was much lower, especially longer than 3 Mb. This finding suggested that ancient and contemporary inbreeding events might impact the Wenchang chicken population, but this population had a low level of inbreeding, and ancient ancestors were the main inbreeding event-affected group [30].

Additionally, some researchers suggested that, in comparison to SNP chips, the higher resolution of whole genome sequence data may lead to the identification of ROH shorter than 1 Mb [33]. In our research, the length of identified ROH segments shorter than 1 Mb was predominant, consistent with previous research on the analysis of other chicken breeds based on whole-genome sequencing data [3]. These short ROH segments may reflect ancestral relationships and more ancient inbreeding events in the Wenchang chicken sample population. In our previous research, the Wenchang chicken sample population displayed a comparatively higher genetic diversity than other chicken breeds; the results of the distribution of ROH segments reconfirmed that finding [11]. At the same time, we verified the influence of different detection parameters on the results. With the increase in the number of single SNPs detected, the number of short fragments detected decreased, but there was almost no effect on the number of long fragments, and the change was not noticeable. Short segments still accounted for the largest proportion of detected SNPs; however, this did not affect the results and conclusions of this experiment.

Each individual’s average length of ROH segments identified across the genome in the Wenchang chicken sample population was 53.53 Mb. This result was much smaller than a previous report on a commercial broiler line, in which the length of ROH was 130.9 Mb on average [33]. Commercial breeds have suffered strong artificial selection pressure for genetic improvement on traits of economic interest. Zhang et al. [34] pointed out that the reason for the difference between Chinese indigenous chicken breeds and commercial breeds on the value of *F_ROH_* might be the difference in selection pressure they have been undergoing. Thus, in comparison with commercial chicken breeds, lower inbreeding events and higher genetic diversity may exist in the Wenchang chicken sample population, consistent with the results of genetic diversity and LD analysis. Moreover, the total length of ROH varied among individuals in the Wenchang chicken sample population, which indicated that ROH differs among individuals, consistent with previous studies [3]. Some scholars have suggested that this difference may be attributed to the lower length threshold used to detect ROH and the higher density of SNPs used to perform analysis [19,34]. 

### 4.3. Inbreeding Coefficients

Estimating inbreeding coefficients based on pedigree data (*F_PED_*) has been widely used in previous studies. However, *F_PED_* may not be considered an accurate estimation of true inbreeding degree because of many limitations, such as errors in pedigree records and the fact that the coefficient does not reflect the random nature of Mendelian sampling and recombination [35]. Many studies have implemented and attested that the estimation of the inbreeding coefficient based on ROH fragments is feasible without pedigree information [7,36,37]. Estimating the inbreeding coefficient based on ROH does not depend on the allele frequencies or the pedigree’s incompleteness. As a result, *F_ROH_* is typically less affected by external factors and is more precise in estimating the degree of inbreeding than other methods [36]. The average value of *F_ROH_* estimated in the Wenchang chicken sample population was 0.0566, and the value of *F_ROH_* (<1 Mb) was significantly larger than others. A similar result was found in other chicken breeds [3]. *F_GRM_* and *F_HOM_* were also calculated in our study to verify the correctness of the inbreeding coefficient of the Wenchang chicken sample population. Comparisons with the values among the inbreeding coefficients calculated by these three methods were similar (approximately 0.05), indicating that the inbreeding degree in the Wenchang chicken sample population was again low. Our results also implied that the estimation of *F_ROH_* based on ROH lengths was reasonably accurate in predicting the genomic inbreeding coefficient, consistent with the conclusion of previous studies [38].

### 4.4. Candidate Genes within ROH Islands

ROH islands may represent regions of the genome that have undergone natural or artificial selection. In our study, 19 genome regions with a high frequency of ROH occurrence were identified. After annotation, we found that some GO terms were related to digestion. For instance, GO:0004556-alpha-amylase activity has been demonstrated to be helpful in the digestion of starch in a corn diet. Some researchers added alpha-amylase to broiler diets and found some influence on the growth rate and the development of digestive organs [39,40]. Hence, these GO terms may relate to the growth and development traits of Wenchang chickens. In addition, the GO:0005518-collagen binding was also enriched in our study. Collagen in muscle is associated with the toughness of meat and can affect the maturation of connective tissue and the tenderness of meat [41]. Thus, this GO term may be associated with meat-quality traits. Wenchang chickens are well known for their juicy and tender meat and are well-received by consumers. The significant KEGG pathway enriched in our study is Extracellular matrix-cell interactions -receptor interaction (GGA04512), which has been reported to play a potentially central role throughout the ovulation cycle [42]. These results could putatively explain some reproduction traits of the Wenchang chicken breed.

For the candidate genes within ROH islands identified in Wenchang chicken, we found some interesting genes that may be related to economically important traits. Among these genes, the *AMY1a* gene has been reported to be strongly associated with growth performance, feed intake, and body shape traits in chickens [43]. Likewise, we found that some genes may influence stress resistance in Wenchang chickens. An example of this would be the *THEMIS2* gene, which has been previously implicated in disease resistance based on whole-genome sequencing data in chickens [44]. The *PIK3C2B* gene has been reported to play an important role in the adaptation mechanisms of ducks to heat stress [45]. Wenchang chickens are produced in the Hainan province of China, where the temperature is relatively high throughout the year and has driven the evolution of strong heat tolerance over time in the local livestock and poultry populations. We also identified some genes related to meat traits in our Wenchang chicken sample population. The *MBTPS1* gene is associated with meat quality parameters such as shearing force [46]. The *DLK1* genes are involved in fat development and differentiation, affecting muscle growth and meat tenderness [47]. *EPS8L2*, a family of eps8-related proteins, is a new protein family responsible for functional redundancy that leads to actin remodeling in RTK-activated signaling pathways. Related research indicates that *EPS8L2* may also play an important role in muscle formation [48]. Fat deposition often has an impact on meat quality and flavor. The *LANCL2* gene, which was enriched in the Wenchang chicken sample populations, was reported to be involved in the process of trans-activation of downstream lipogenic genes mediated by *PPARγ* [49]. Thus, this gene may be linked to fat deposition and indirectly affect the meat quality trait of Wenchang chickens. In brief, the regions identified in this study may help explain the genetic mechanisms underlying the favorable qualities of Wenchang chickens.

## 5. Conclusions

In summary, in this study, we detected the ROH across the genome of a Wenchang chicken sample population and calculated the inbreeding coefficient to investigate the degree of inbreeding. We also identified the candidate regions within ROH islands that contain genes related to the economically important and identifying characteristics of Wenchang chickens. Our findings demonstrated that historical inbreeding events had little impact on the Wenchang chicken sample population, which displayed a relatively low level of inbreeding. Based on the enrichment analysis of identified candidate regions within ROH islands, we found some genes were related to the economically important traits of Wenchang chicken, such as body shape, meat quality, disease resistance, and heat tolerance. Overall, our research provides evidence for a better understanding of the genetic mechanisms controlling Wenchang chicken characteristics and provides insight into inbreeding events for preservation strategies and utilizing Wenchang chickens in the future.

## Figures and Tables

**Figure 1 animals-13-01645-f001:**
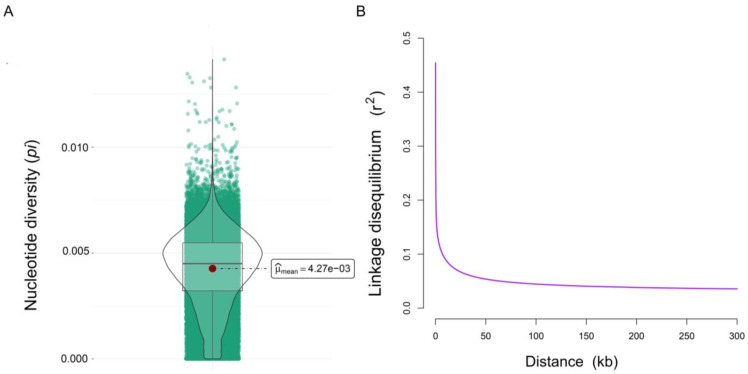
The nucleotide diversity (**A**) and linkage disequilibrium decay (**B**) of Wenchang chickens. (**A**) The violin plot of nucleotide diversity (*pi*) of Wenchang chickens, where the red dot represents the average value. (**B**) The linkage disequilibrium decay of Wenchang chickens. The x-axis represents SNP marker distance (kb), and the y-axis is the squared correlation (*r*^2^) between pairwise SNPs.

**Figure 2 animals-13-01645-f002:**
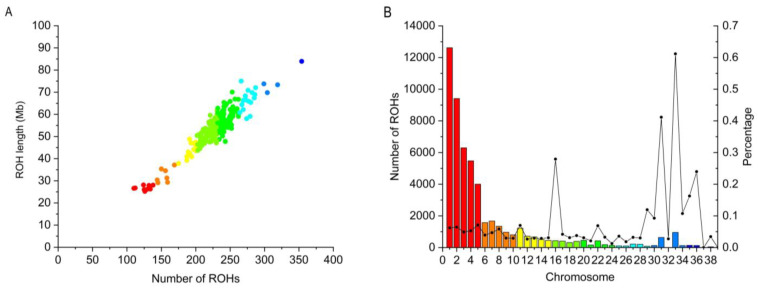
The distribution of ROH detected in Wenchang chicken across autosomes. (**A**) Length distribution of ROH. The x-axis represents the number of ROH, and the y-axis represents the length of ROH (Mb). (**B**) Number and coverage of the ROH. The bars and the lines represent the number of ROH and ROH coverage, respectively, on each chromosome.

**Figure 3 animals-13-01645-f003:**
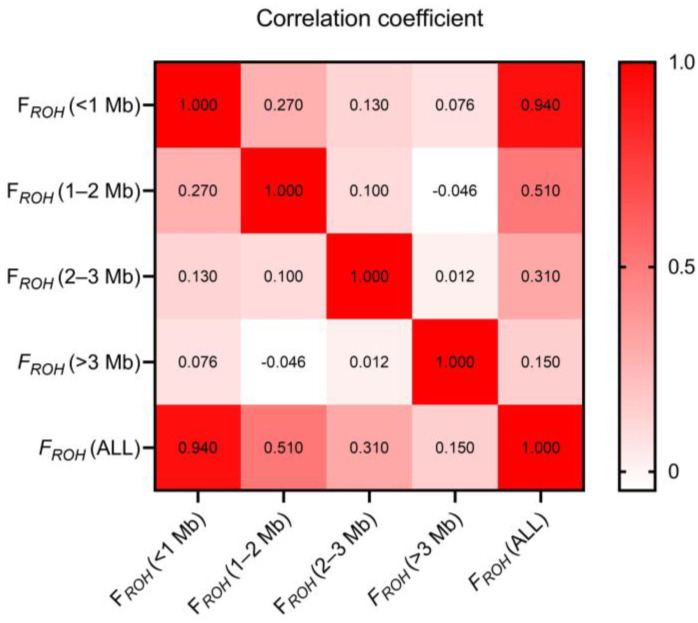
Correlation of genomic inbreeding coefficients calculated based on different length ROH fragments (*F_ROH_* (<1 Mb), *F_ROH_* (1–2 Mb), *F_ROH_* (2–3 Mb), *F_ROH_* (>3 Mb), and *F_ROH_* (All)). The darker the color, the higher the correlation of the data. The *t*-test of all the data shows that the *p*-value between the data is less than 0.01.

**Figure 4 animals-13-01645-f004:**
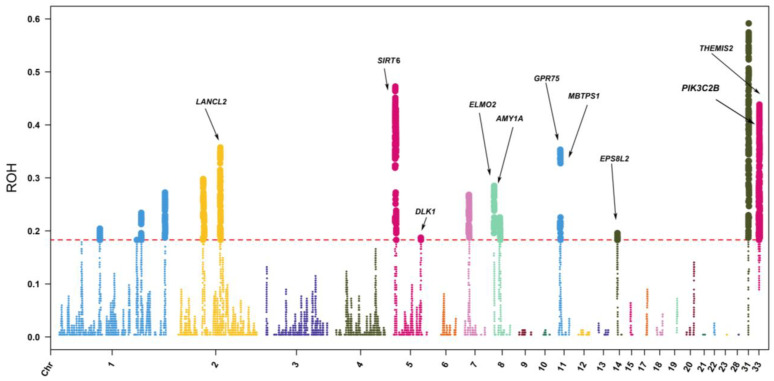
Manhattan plot of the frequency for each SNP within ROH regions among all individuals. The horizontal red line indicates the threshold for top 1%. The x-axis represents positions along each chromosome. Genes related to reproduction quality are identified.

**Figure 5 animals-13-01645-f005:**
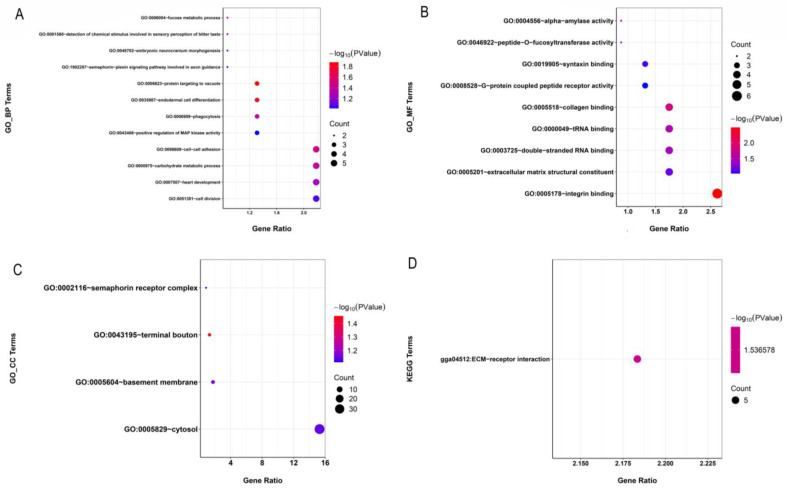
Enrichment analysis of the variants located in gene exons. The significantly enriched GO terms (*p* < 0.05) are classified as a biological process (BP, **A**), cellular component (CC, **B**), or molecular function (MF, **C**). The significantly enriched KEGG pathways (*p* < 0.05) are shown in (**D**). The size of each circle represents the number of genes in each GO term or pathway, and the color represents the *p*-value of each GO term or pathway.

**Table 1 animals-13-01645-t001:** Genetic diversity indices of Wenchang chickens.

H_E_	H_O_	*P_N_*	MAF	*pi*
0.241613 ± 0.162092	0.228047 ± 0.154646	0.8264	0.170253 ± 0.143041	0.004268 ± 0.001726

**Table 2 animals-13-01645-t002:** Descriptive statistics of runs of homozygosity (ROH) numbers and lengths (in Mb) by ROH length classes (0–1 Mb, 1–2 Mb, 2–3 Mb, ROH > 3 Mb and total).

ROH Length (Mb)	ROH Number	Percent (%)	Mean Length (Mb)	Genome Coverage (%)
0–1	52,526	98.17%	0.212 ± 0.147	88.53%
1–2	853	1.59%	1.323 ± 0.264	8.97%
2–3	111	0.21%	2.302 ± 0.259	2.03%
>3	16	0.03%	3.64 ± 0.600	0.46%
Total	53,506	100.00%	0.235 ± 0.232	100.00%

**Table 3 animals-13-01645-t003:** The average genomic inbreeding coefficient of *F_HOM_*, *F_GRM_*, and *F_ROH_* for different length categories of ROH.

*F_ROH_* (<1 MB)	*F_ROH_* (1–2 MB)	*F_ROH_* (2–3 MB)	*F_ROH_* (>3 MB)	*F_ROH_* (ALL)	*F_HOM_*	*F_GRM_*
0.0501 ± 0.0084	0.0051 ± 0.0027	0.0012 ± 0.0017	0.0003 ± 0.001	0.0566 ± 0.0100	0.0561 ± 0.0205	0.0600 ± 0.0144

## Data Availability

The raw data used in this study are publicly available and can be obtained upon reasonable request to the corresponding author.

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
