# Peer review of "Identification of Runs of Homozygosity Islands and Functional Variants in Wenchang Chicken"

_animals, 2023, doi:10.3390/ani13101645_

Round 1

Reviewer 1 Report

The manuscript by Thian et al. is reporting on runs of homozygosity and inbreeding of Wenchang chicken from Hainan Province. Whole genome sequencing of in total 235 chicken has been performed to elucidate the genomic characteristics of this chicken breed. The manuscript is interesting and original.

However, I am concerned about the ROH analysis approach. The authors are using a rule-based approach for ROH calling, which requires thorough parameter adjustments specifically for the analyzed dataset. The parameters suggested as default by PLINK are not sufficient for whole genome sequencing analysis. The authors should check recent publications providing formula for estimating parameters, e.g. Purefield, 2013 (https://link.springer.com/article/10.1186/1471-2156-13-70). Based on the chicken genome size and the number of called SNPs I would expect a density of about 65 to be the best fitting solution. Otherwise the minimal length of a ROH of 5000kb (100kb*50 SNPs) is apparently too long. The issue of different parameter settings should also be discussed (seel also Berghöfer, 2022 https://link.springer.com/article/10.1186/s12864-015-1977-3).  This might also be relevant when comparing the results with data from other studies as mentioned in the discussion in section 4.2 (commercial broiler line). Where the same parameters applied?

Furthermore, it is unclear if Chromosomes W and Z were considered in the ROH analysis (they shouldn’t be included or at least considered separately). What about Chromosome 39? I assume no ROHs were detected according to the small genome size?

Minor comments:

Introduction: remove blank:…breeding may also increase the probability of genetic drift…

Introduction: Besides consanguineous mating…artificial selection of animals..

Author Response

Comments and Suggestions for Authors

The manuscript by Thian et al. is reporting on runs of homozygosity and inbreeding of Wenchang chicken from Hainan Province. Whole genome sequencing of in total 235 chicken has been performed to elucidate the genomic characteristics of this chicken breed. The manuscript is interesting and original.

However, I am concerned about the ROH analysis approach. The authors are using a rule-based approach for ROH calling, which requires thorough parameter adjustments specifically for the analyzed dataset. The parameters suggested as default by PLINK are not sufficient for whole genome sequencing analysis. The authors should check recent publications providing formula for estimating parameters, e.g. Purefield, 2013 (https://link.springer.com/article/10.1186/1471-2156-13-70).

Response:

Many thanks for your comments. The parameters we used to detect the ROH across the autosome was refered to some ROH studies on chickens(Cendron et al., 2020, Poultry Science; Zhang et al., 2018, BMC Genomics). Specific parameters are as follows:(1) Each sliding window should contain 50 SNPs across the genome; (2) due to genotyping error, up to five SNPs with missing genotypes and one SNP with a heterozygous genotype were allowed for each ROH; (3) each ROH should have a sequence of more than 50 consecutive SNPs; and (4) only detect segments with ROH length greater than 100 kilobases (kb). ROHs extracted from sequence data were further classified into three length categories: short ROHs [<1 megabase (Mb)], medium ROHs (1-2 Mb and 2-3 Mb), and long ROHs (> 3 Mb). In addition, to further understand the influence of different parameters, we use the detection method of adding 50 SNPs at a time to verify the influence of different parameters. See Line 123-133.

Based on the chicken genome size and the number of called SNPs I would expect a density of about 65 to be the best fitting solution. Otherwise the minimal length of a ROH of 5000kb (100kb*50 SNPs) is apparently too long.

Response:

Many thanks for your comments. I am sorry that we may have made a mistake in describing the ROH detection parameters. The minimum length is 100kb instead of 5000kb, we have modified in Line 128 in the revision.

The issue of different parameter settings should also be discussed (seel also Berghöfer, 2022 https://link.springer.com/article/10.1186/s12864-015-1977-3).  This might also be relevant when comparing the results with data from other studies as mentioned in the discussion in section 4.2 (commercial broiler line). Where the same parameters applied?

Response: Many thanks for your comments. We have added the analysis of the the same parameters applied according your suggestion. The results have been added in TableS4. Details see Line 211-214 and Line 331-337.

Furthermore, it is unclear if Chromosomes W and Z were considered in the ROH analysis (they shouldn’t be included or at least considered separately). What about Chromosome 39? I assume no ROHs were detected according to the small genome size?

Response: Many thanks for your comments. In this study, we only detected the ROHs across the autosome, the chromosomes W and Z were not considered. It has been described in the reversed manuscript, see Line 122. Besides, no ROH has been detected in chromosome 39 due to the small genome size.

Minor comments:

Introduction: remove blank:…breeding may also increase the probability of genetic drift…

Response: Many thanks for your comments. I have deleted the blanks in the reversion.

Introduction: Besides consanguineous mating…artificial selection of animals..

Response: Many thanks for your comments. I have deleted the blanks in the reversion.

Reviewer 2 Report

In this article entitled "Identification of Runs of Homozygosity Islands and Functional Variants in Wenchang Chicken,", Tian et al. provided valuable insights into the genomic characteristics of Wenchang chicken, particularly with regards to the identification of Runs of Homozygosity (ROH) islands and functional variants. The paper is interesting, the experimental approach is appropriate, and I commend the authors for their research efforts. However, I have noticed that the article has some serious problems with English grammar and writing. While I understand that English may not be your first language, the quality of the writing is essential for communicating your research findings effectively. The errors in the manuscript can make it challenging for readers to understand your research methodology, results, and conclusions. I suggest that you seek assistance from a professional editor or a language editing service to improve the language quality of your manuscript. The authors need to incorporate some minor changes, as mentioned below:

·       In line 11: the sentence: “and also the only 10 chicken breed listed in the‘animal genetic resources in China (poultry)’in Hainan Province.”  is poorly structured and hard to understand

·       In line 93: (also elsewhere in the manuscript) please remove “by”

·       In line 94: please rewrite this sentence: “the details see our previous studies and the sequencing information see Table S1”

·       In line 96:  please remove “the”.

·       In line 143: please correct the word “first”.

·       In line 168: please replace “was” by “is” and correct the word “value”.

·       In section 3.1: please replace “chromosomes 6” by “chromosome 6”

·       In Figure 1 and 4 The labels should be added to indicate what the authors are describing

·        The labels in Figure 3 are difficult to read, please improve it

Author Response

Comments and Suggestions for Authors

In this article entitled "Identification of Runs of Homozygosity Islands and Functional Variants in Wenchang Chicken,", Tian et al. provided valuable insights into the genomic characteristics of Wenchang chicken, particularly with regards to the identification of Runs of Homozygosity (ROH) islands and functional variants. The paper is interesting, the experimental approach is appropriate, and I commend the authors for their research efforts. However, I have noticed that the article has some serious problems with English grammar and writing. While I understand that English may not be your first language, the quality of the writing is essential for communicating your research findings effectively. The errors in the manuscript can make it challenging for readers to understand your research methodology, results, and conclusions. I suggest that you seek assistance from a professional editor or a language editing service to improve the language quality of your manuscript.

Response: Many thanks for your comments. The English of this manuscript has already been edited by a professional language editing service, and you can find the certificate below.

The authors need to incorporate some minor changes, as mentioned below:

In line 11: the sentence: “and also the only 10 chicken breed listed in the‘animal genetic resources in China (poultry)’in Hainan Province.”  is poorly structured and hard to understand

Response: Many thanks for your comments. The sentence have been replaced by “Wenchang chickens are the only chicken breed listed in the ‘animal genetic resources in China (poultry)’ in Hainan Province and are famous for their excellent meat quality. See Line 13-14.

In line 93: (also elsewhere in the manuscript) please remove “by”

Response: Many thanks for your comments. The “by” has been removed and the English of this manuscript has already been edited by a professional language editing service.

In line 94: please rewrite this sentence: “the details see our previous studies and the sequencing information see Table S1”

Response: Many thanks for your comments. The sentence have been replaced by “ All selected individuals’ genomes were sequenced using the Illumina Nova Seq platform (Illumina, San Diego, CA, USA) and 150-base pair (bp) paired-end sequencing. For details, see Table S1”, see Line 99-101.

In line 96:  please remove “the”.

Response: Many thanks for your comments. We have removed "the".

In line 143: please correct the word “first”.

Response: Many thanks for your comments. We have correct the word “first”, see Line 154.

In line 168: please replace “was” by “is” and correct the word “value”.

Response: Many thanks for your comments. The misspell has been corrected, and the English of this manuscript has already been improved by native speakers.

In section 3.1: please replace “chromosomes 6” by “chromosome 6”

Response: Many thanks for your comments. We have replace “chromosomes 6” by “chromosome 6”, see line 176.

In Figure 1 and 4 The labels should be added to indicate what the authors are describing

Response: Many thanks for your comments. We have added the lables of Figure 1 and 4 in the reversion, see Line 192-197, Line 253-255.

The labels in Figure 3 are difficult to read, please improve it

Response: Many thanks for your comments. To make it easier to understand, we have drawn a heat map to visible results in the reversion, see the new Figure 3.

Round 2

Reviewer 1 Report

The manuscript has been significantly improved by the revisions. I suggest publication in its current state.